# Secrecy Energy Efficiency Maximization for Secure Unmanned-Aerial-Vehicle-Assisted Simultaneous Wireless Information and Power Transfer Systems

**Daehan Ha** [1] , **Seongah Jeong** [2] , **Jinkyu Kang** [3] **and Joonhyuk Kang** [1,*]

1   School of Electrical Engineering, Korea Advanced Institute of Science and Technology (KAIST), Daejeon 34141, Republic of Korea; daehan.ha@kaist.ac.kr
2   School of Electronics Engineering, Kyungpook National University, Daegu 14566, Republic of Korea; seongah@knu.ac.kr
3   Department of Information and Communication Engineering, Myongji University, Yongin-si 17058, Republic of Korea; jkkang@mju.ac.kr
*   Correspondence: jhkang@ee.kaist.ac.kr; Tel.: +82-42-350-7422

**Abstract:** Unmanned aerial vehicle (UAV)-assisted simultaneous wireless information and power transfer (SWIPT) systems have recently gained significant attraction in internet-of-things (IoT) applications that have limited or no infrastructure. Specifically, the free mobility of UAVs in three-dimensional (3D) space allows us good-quality channel links, thereby enhancing the communication environment and improving performance in terms of achievable rates, latency, and energy efficiency. Meanwhile, IoT devices can extend their battery life by harvesting the energy following the SWIPT protocol, which leads to an increase in the overall system lifespan. In this paper, we propose a secure UAV-assisted SWIPT system designed to optimize the secrecy energy efficiency (SEE) of a ground network, wherein a base station (BS) transmits confidential messages to an energy-constrained device in the presence of a passive eavesdropper. Here, we employ a UAV acting as a helper node to improve the SEE of the system and to aid in the energy harvesting (EH) of the battery-limited ground device following the SWIPT protocol. To this end, we formulate the SEE maximization problem by jointly optimizing the transmit powers of the BS and UAV, the power-splitting ratio for EH operations, and the UAV's flight path. The solution is obtained via a proposed algorithm that leverages successive convex approximation (SCA) and Dinkelbach's method. Through simulations, we corroborate the feasibility and effectiveness of the proposed algorithm compared to conventional partial optimization approaches.

**Keywords:** unmanned aerial vehicle (UAV); physical-layer security (PLS); secrecy energy efficiency (SEE); simultaneous wireless information and power transfer (SWIPT)





## 1. Introduction

Unmanned aerial vehicles (UAVs) have recently garnered attention for internet-of-things (IoT) networks due to their high mobility, affordable installation costs, and flexibility in deployment. These advantages of UAVs are particularly pronounced in environments with limited or no infrastructure, such as rural areas, disaster and emergency response scenarios, military services, advanced radio technology, etc. [1–3]. However, the constrained energy capacity and limited coverage range of energy-restricted UAVs introduce new challenges in the design of wireless communication systems. Consequently, recent research efforts in UAV-assisted wireless networks have pivoted towards addressing energy efficiency and energy-aware deployment issues.

For energy-constrained devices, radio frequency wireless power transfer (WPT) technology emerges as an effective and attractive solution. In UAV-assisted IoT networks, the advantages of WPT are further highlighted as devices harvest energy from radio signals, thereby recharging battery-limited devices, i.e., IoT devices and UAVs, and extending

the overall system lifespan. To actualize UAV-assisted WPT in IoT environments, several architectures have been explored, such as wireless powered communication networks (WPCN) and simultaneous wireless information and power transfer (SWIPT). In subsequent sections, we delve into the related works concerning UAV-assisted networks and WPT in greater detail [4–17].

### 1.1. Related Work

### 1.1.1. UAV-Assisted Networks

As UAVs possess the ability to fly freely within their operational capabilities, they are well suited for enhancing physical-layer secrecy [4]. In studies such as [5–11], UAV-assisted network systems have been explored, wherein the UAV serves as a relay to facilitate communication between wireless devices. For instance, UAVs can operate as helper nodes, moving away from potential eavesdroppers (Eves or Es) while orbiting the intended user to transmit confidential information. Conversely, as jammers, UAVs can hover near Eves to cause interference while maintaining a distance from the intended user to avoid disrupting the signal. For this reason, the UAVs have been actively explored [5–8] to improve physical-layer security. Compared to the prior research in the field of physical-layer security, the study in [5] represents the first instance where enhancements in the secrecy rate have been achieved via jointly optimizing power control and trajectory under the assumption that the location of the Eve is perfectly known. In contrast, the influence of imprecise information about multiple Eves' locations on the UAV's trajectory and transmit power design is examined in [6]. The authors in [7] propose the full-duplex secrecy communication scheme for the UAV to achieve the maximum secrecy energy efficiency (SEE) of the UAV. Furthermore, ref. [8] describes a UAV-enabled data collection system where an intelligent reflecting surface (IRS) aids communication between a cluster of IoT devices and a UAV, despite the presence of a malicious jammer.

Regarding IoT systems with battery-limited devices, several studies [9–11] focus on the energy consumption or energy efficiency of IoT devices with UAV support. The authors in [9] propose the joint optimization problem of resource allocation and UAV's path planning with the aim of minimizing the end devices' energy consumption for the mobile edge computing system via a UAV-mounted cloudlet. In [10], the minimization of energy consumption is addressed in the rotary-wing UAV-assisted communication system. The design of maximizing the energy efficiency (EE) through the UAV's trajectory optimization is studied in [11]. Nevertheless, considering that UAVs are also constrained by their battery life, which limits their available operation time, their finite energy budget must be considered in the system design [4–11].

### 1.1.2. Wireless Power Transfer (WPT)

The concept of UAV-enabled WPT was introduced in [12,13], wherein UAVs act as mobile energy transmitters to recharge low-power devices on the ground. By leveraging their free mobility, UAVs can strategically position themselves to minimize the distance to targeted ground users, thereby enhancing the efficiency of both wireless information transfer (WIT) and WPT. The work in [14] addresses the throughput maximization problem in a UAV-based WPCN by jointly optimizing the UAV's trajectory and the allocation of transmission resources for both uplink WIT and downlink WPT. Moreover, the integration of UAV-assisted communication with SWIPT is proposed in [15] to facilitate IoT networks during emergency situations. In [16], the researchers investigate the maximization of the secrecy rate in a UAV-enabled communication network that conducts SWIPT in a millimeter-wave (mmWave) communication environment suitable for IoT applications. The paper [17] discusses the novel design challenges and strategies for UAV-assisted wireless energy transfer in anticipation of the forthcoming 6G era, which will be characterized by the internet of everything. Prior works such as [14–17] focus on user throughput or secrecy rates in UAV-assisted SWIPT systems, taking into account the UAV's constrained energy but not from an EE perspective.

*1.2. Our Contributions*

Motivated by the limitations of previous works [4–17] in terms of secrecy and EE, we investigate the SEE for secure UAV-assisted SWIPT systems. SEE is defined as the ratio between the achievable secrecy rate of the desired link and the UAV's energy consumption. In our setup, a base station (referred to as Alice or A) communicates confidentially with an energy-constrained IoT device (referred to as Bob or B) on the ground, with the assistance of a UAV and in the presence of a passive Eve. To maximize the SEE of the system, we formulate an optimization problem and develop a corresponding algorithmic solution. The main contributions of this paper are summarized as follows:

- We formulate the SEE maximization problem, jointly optimizing Alice's transmit power, the UAV's transmit power, Bob's power splitting ratio for energy harvesting (EH), and the UAV's trajectory, considering the functional capability constraints of network devices, such as maximum speed, average power, and peak power.
- To solve the formulated problem, we propose an efficient iterative algorithm based on successive convex approximation (SCA) [18] and Dinkelbach's method [19], which converges to a local minimum of the original non-convex problem.
- Through simulations, we demonstrate the superior performance of proposed algorithm compared to conventional partial optimization methods. To the best of our knowledge, this is the first study to focus on the SEE of UAV-assisted communication systems with EH in an energy-constrained environment.

The remainder of this paper is organized as follows: Section 2 describes the system model and performance metrics. Section 3 details the formulation of the SEE maximization problem and the development of the algorithm for finding local optimal solutions. Numerical results and concluding remarks are presented in Sections 4 and 5, respectively.

## 2. System Model and Performance Metric

We consider a secure UAV-assisted SWIPT system, as shown in Figure 1. Alice transmits the confidential information to Bob with the aid of a rotary-wing UAV, while a passive Eve attempts to wiretap it. Without loss of generality, a three-dimensional Cartesian coordinate system is adopted, whose coordinates are measured in meters. We assume that Alice, Bob, and Eve are located at the position $\{L_i = (x_i, y_i, 0)$, for $i \in \{A, B, E\}$, in the xy-plane, e.g., on the ground, while the UAV flies along a trajectory $L_U(t) = (x_U(t), y_U(t), H)$ with the fixed altitude of $H$, for the given flying period $0 \leq t \leq T$. Due to the horizontally constant altitude $H$ for the UAV, we focus on the projection of the UAV's trajectory onto the xy-plane. Additionally, for the aerospace regulations and the operational capability, the initial and final positions and the maximum velocity of the UAV are predetermined as $L_{U_I} = (x_{U_I}, y_{U_I}, H)$, $L_{U_F} = (x_{U_F}, y_{U_F}, H)$ and $v_{\max}$, respectively. The UAV is assumed to operate in a time division duplex (TDD) manner. Alice transmits the confidential messages to the UAV, and then the UAV forwards the decoded messages to Bob with the attempt to improve the secrecy rate while supporting the EH operation of Bob. Here, by following the SWIPT protocol design in [16], the received signal power at Bob is split into two types of power streams such as one portion $w$ of the power for data communication and the remaining portion $(1 - w)$ of the power for EH with satisfying $0 \leq w \leq 1$.

We consider that the mission time $T$ is discretized into $N$ time slots with a sufficiently small time $t_s = T/N$ in seconds (s) [7,9]. Accordingly, the UAV's position $L_U(t)$ can be discretized as $L_U[n] = (x_U[n], y_U[n], H)$, for $n \in \mathcal{N} = \{1, \cdots, N\}$. In addition, the distance $d_s = v_{\max} t_s$ denotes the maximum distance of horizontally traveling within each time slot with the maximum speed $v_{\max}$ in m/s. The distance of $d_s$ is chosen to be sufficiently small compared to $H$ so that the air-to-ground channels of the UAV are assumed to be time-invariant within each slot. We define $L_U[n] = (x_U[n], y_U[n], H)$ as the location of the UAV for all time slot $n$, which satisfies the mobility constraints as follows:

$$\|L_U[n] - L_U[n-1]\| \leq d_s \text{ for } n \in \mathcal{N}, \tag{1}$$

$$L_U[0] = L_{U_I} \text{ and } L_U[N] = L_{U_F}. \tag{2}$$

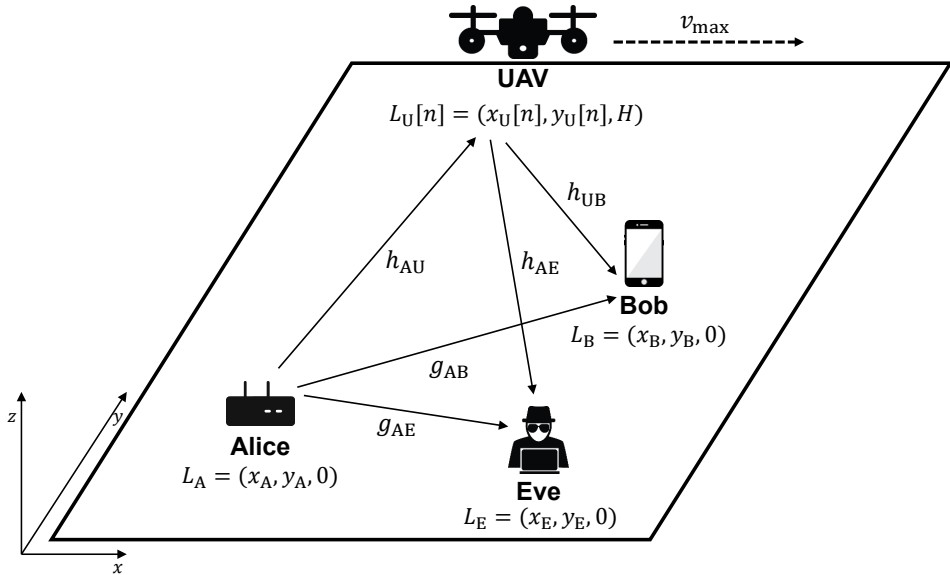

**Figure 1.** Illustration of the secure UAV-assisted SWIPT systems.

Following [7,9,16], the air-to-ground channels between the UAV and ground nodes, i.e., Alice, Bob, and Eve, are assumed to be dominated by line-of-sight (LOS) links. At the $n$-th slot, the channel gain between the UAV and ground nodes is given by

$$h_{\mathrm{U}j}[n] = \rho_0 (d_{\mathrm{U}j}[n])^{-\alpha} = \rho_0 (\|\boldsymbol{L}_{\mathrm{U}}[n] - \boldsymbol{L}_j\|^2 + H^2)^{-\frac{\alpha}{2}} \quad \text{for } j \in \{A, B, E\}, \tag{3}$$

where $d_{ij}[n] = (\|\boldsymbol{L}_i[n] - \boldsymbol{L}_j\|^2 + H^2)^{1/2}$ is the Euclidean distance between node $i$ and $j$, $\alpha$ denotes the pathloss exponent of the channels, and $\rho_0$ denotes the gain of a channel with the reference distance $d_0 = 1$m. The ground channels from Alice to Bob and Eve are assumed to be Rayleigh channels. Hence, the channel coefficients from Alice to Bob and Eve in slot $n$ are given by $g_{\mathrm{A}j}[n] = \rho_0 (d_{\mathrm{A}j}[n])^{-\alpha} \zeta_j$ for $j \in \{B, E\}$, where $\alpha$ denotes the pathloss exponent and $\zeta_j$ is an exponentially distributed random variable with unit mean accounting for the small-scale Rayleigh fading. Since $t_s$ is assumed to be much smaller than the coherence time of the ground channels, the channels are supposed to be stationary and ergodic within each time slot.

We define $P_{\mathrm{A}}[n]$ and $P_{\mathrm{U}}[n]$ as the transmit powers of Alice and UAV at time slot $n$, respectively, which, in real applications, need to satisfy the average and peak power constraints as [5]:

$$\frac{1}{N} \sum_{n=1}^{N} P_{\mathrm{j}}[n] \leq P_{\mathrm{j}}^{\mathrm{avg}}, \quad 0 \leq P_{\mathrm{j}}[n] \leq P_{\mathrm{j}}^{\mathrm{peak}} \quad \text{for } n \in \mathcal{N}, \tag{4}$$

where $P_{\mathrm{j}}^{\mathrm{avg}} \leq P_{\mathrm{j}}^{\mathrm{peak}}$ for $j \in \{A, U\}$. Also, it is noted that the communication energy consumption is much smaller than the propulsion energy, which allows us to omit the communication energy consumption in the following SEE calculation [7,11]. Based on [10], the UAV's propulsion energy consumption $E_{\mathrm{P}}[n]$ at time slot $n$, consisting of blade profile, induced power, and parasite power in Joules (J), can be modeled as

$$E_{\mathrm{P}}[n] = t_s \left( P_0 \phi[n] + P_i (\psi[n])^{1/2} + \frac{1}{2} d_0 \rho s A (v[n])^3 \right), \tag{5}$$

where the horizontal UAV's flying speed is given by $v[n] = \|\boldsymbol{L}_{\mathrm{U}}[n] - \boldsymbol{L}_{\mathrm{U}}[n-1]\|/t_s$; $\phi[n] = 1 + 3(v[n])^2/U_{\mathrm{tip}}^2$ with $U_{\mathrm{tip}}$ being the tip speed of the UAV's rotor blade; $\psi[n] = (1 + (v[n])^4/(4v_0^4))^{1/2} - (v[n])^2/(2v_0^2)$; $P_0$, $P_i$ and $v_0$ represent the blade profile power, the induced power, and the mean rotor-induced speed when the UAV is hovering, re-

spectively; and $d_0$, $\rho$, $s$, and $A$ denote the fuselage drag ratio, the air density, the rotor solidity, and the rotor disc area, respectively. Note that (5) is practically valid for the straight and level flight of the rotary-wing UAV, which is satisfied in each time slot $n$ due to the approximated piecewise-linear trajectory over time slots.

Under the assumption that the Eve adopts the maximal ratio combining (MRC) scheme to intercept the confidential information, the secrecy rate $R_S[n]$ between Alice and Bob can be expressed as [20]

$$R_S[n] = [R_{AB}[n] - R_{AE}[n]]^+, \tag{6}$$

where $[x]^+ \triangleq \max(x, 0)$;

$$R_{AB}[n] = \min(\log_2(1 + \gamma_{AU}[n]), \log_2(1 + \gamma_{AB}[n] + \gamma_{UB}[n])) \tag{7}$$

with $\gamma_{AU}[n] = \gamma_{AU}(P_A[n], \boldsymbol{L}_U[n]) = h_{AU}[n]P_A[n]/\sigma_U^2$, $\gamma_{AB}[n] = \gamma_{AB}(P_A[n], w) = wg_{AB}$ $P_A[n]/\sigma_B^2$ and $\gamma_{UB}[n] = \gamma_{UB}(P_U[n], w, \boldsymbol{L}_U[n]) = wh_{UB}[n]P_U[n]/\sigma_B^2$ representing a signal-to-noise ratio (SNR) of Alice to Bob, Alice to UAV, and UAV to Bob, respectively;

$$R_{AE}[n] = \log_2(1 + \gamma_{AE}[n] + \gamma_{UE}[n]) \tag{8}$$

with $\gamma_{AE}[n] = \gamma_{AE}(P_A[n]) = g_{AE}P_A[n]/\sigma_E^2$ and $\gamma_{UE}[n] = \gamma_{UE}(P_U[n], \boldsymbol{L}_U[n]) = h_{UE}[n]P_U[n]$ $/\sigma_E^2$ representing the SNR of Alice to Eve and UAV to Eve, respectively, and $\sigma_B^2$, $\sigma_U^2$, and $\sigma_E^2$ are the noise power at Bob, UAV, and Eve, respectively.

The goal of this paper is to maximize the SEE of the UAV-assisted SWIPT systems under the transmit power constraints of Alice and UAV, the EH constraint of Bob, and the flying constraints for UAV. In wireless communications, SEE is a pivotal metric that quantifies how much confidential data a system can relay per energy unit used, capturing both its energy efficiency and secure transmission proficiency [7]. To this end, we define the SEE in bits/J that concerns the technical ratio between the secrecy rate $R_S[n]$ in (6) and the UAV's flying energy consumption in (5) as

$$\text{SEE}(\boldsymbol{P}_A, \boldsymbol{P}_U, w, \boldsymbol{L}_U) = \frac{B \sum_{n=1}^N R_S[n]}{\sum_{n=1}^N E_P[n]}, \tag{9}$$

with Alice's transmit power $\boldsymbol{P}_A \triangleq \{P_A[n]\}_{n=1}^N$, the UAV's power $\boldsymbol{P}_U \triangleq \{P_U[n]\}_{n=1}^N$, power splitting ratio $w$, UAV's trajectory $\boldsymbol{L}_U \triangleq \{\boldsymbol{L}_U[n]\}_{n=1}^N$, where $B$ denotes the system bandwidth. In (9), it is noted that the flying energy consumption of UAV dominates the other energy consumption, e.g., communication, computing, etc. [10].

## 3. Secrecy Energy Efficiency Maximization

In this paper, we aim to maximize the SEE in (9) for the secure UAV-assisted SWIPT communication systems over $N$ time slots by jointly optimizing Alice's transmit power, the UAV's transmit power, the UAV's trajectory, and Bob's power splitting ratio for EH operation. To this end, we formulate the optimization problem as

$$\max_{\boldsymbol{P}_A, \boldsymbol{P}_U, w, \boldsymbol{L}_U} \text{SEE}(\boldsymbol{P}_A, \boldsymbol{P}_U, w, \boldsymbol{L}_U), \tag{10}$$

$$\text{s.t.} \quad \frac{1}{N} \sum_{n=1}^N (g_{AB}P_A[n] + h_{UB}[n]P_U[n])(1 - w)\eta \geq \varphi, \tag{11}$$

$$(1), (2) \text{ and } (4), \tag{12}$$

where (11) represents the EH constraint with $\eta$ being energy conversion efficiency and $\varphi$ being power threshold of Bob. The problem (10) is not convex since the objective function (10) is jointly concave with respect to the optimization variables. To resolve the problem (10), we develop an efficient iterative Algorithm 1 to obtain a high-quality local optimal solution of problem (10). In particular, in each iteration of the proposed

Algorithm 1, we divide the original problem (10) into three subproblems; that is, (1) one optimizes $P_A$ and $P_U$ with fixed $w$ and $L_U$; (2) another optimizes $w$ with fixed $P_A$, $P_U$, and $L_U$; and (3) the other optimizes $L_U$ with fixed $P_A$, $P_U$, and $w$. In the following, we detail the procedure to find the solution of each subproblem per iteration of Algorithm 1.

---

**Algorithm 1:** SEE Maximization Algorithm

---

1 **Initialization**: Input $P_A^0$, $P_U^0$, $w^0$, and $L_U^0$. Let $k = 0$.
2 **Repeat**
3    1. With $w^k$ and $L_U^k$, update $P_A^{k+1}$ and $P_U^{k+1}$ by (18);
4    2. With $P_A^{k+1}$, $P_U^{k+1}$, and $L_U^k$, update $w^{k+1}$ by (22);
5    3. With $P_A^{k+1}$, $P_U^{k+1}$, and $w^{k+1}$, update $L_U^{k+1}$ by (40);
6    **Repeat** (Dinkelbach's method) [19]:
7       Set the Numerator and Denominator of (40) as $\Xi(x,y)$ and $Y(x,y)$;
8       Step1. Set $\lambda_1 = \Xi(\dot{x},\dot{y})/Y(\dot{x},\dot{y})$ for arbitrary $\dot{x} \in \mathcal{X}, \dot{y} \in \mathcal{Y}$;
9       Step2. Formulate $D(x_i,y_i) = \max\{\Xi(x,y) - \lambda_i Y(x,y)|x \in \mathcal{X}, y \in \mathcal{Y}\}$.
10         Solve linear program $D(x_i,y_i)$. Denote the solution as $x_i$ and $y_i$;
11       Step3. **If** $D(x_i,y_i) < \delta$ for $\delta(k) \in (0,1]$, $\delta(k) \to 0$, and $\sum_k \delta(k) = \infty$, stop;
12         **Else** Let $\lambda_{i+1} = \Xi(x_i,y_i)/Y(x_i,y_i)$. Go to Step 2 to replace $i$ with $i+1$;
13    4. Set $\text{SEE}_{k+1}(P_A^{k+1}, P_U^{k+1}, w^{k+1}, L_U^{k+1})$, $k \leftarrow k+1$;
14 **until** (convergence criterion is satisfied)

---

### 3.1. Optimization of Transmit Powers

Here, we optimize the transmit powers of Alice and the UAV with the fixed $w$ and $L_U$. By adopting the slack variable $S = \{S[n]\}_{n=1}^N$ for (10), the SEE problem (10) can be reformulated as

$$\max_{P_A, P_U, S} \sum_{n=1}^N S[n], \tag{13}$$

$$\text{s.t.} \quad S[n] \leq \frac{B}{2}\{\log_2(1 + a_1 P_A[n]) - \log_2(1 + a_3 P_A[n] + a_4[n] P_U[n])\} \quad \text{for} \ n \in \mathcal{N}, \tag{14}$$

$$S[n] \leq \frac{B}{2}\{\log_2(1 + a_2 P_A[n] + a_5[n] P_U[n]) - \log_2(1 + a_3 P_A[n] + a_4[n] P_U[n])\}$$
$$\text{for} \ n \in \mathcal{N}, \tag{15}$$

$$(4) \text{ and } (11), \tag{16}$$

where we assume that the flying energy consumption of $E_P$ only depends on the velocity $v$, and we define $a_1 = h_{AU}[n]/\sigma_U^2$, $a_2[n] = w g_{AB}/\sigma_B^2$, $a_3 = g_{AE}/\sigma_E^2$, $a_4[n] = h_{UE}[n]/\sigma_E^2$, and $a_5[n] = w h_{UB}[n]/\sigma_B^2$. Since the problem (13) with the constraints (14) and (15) involving difference of convex (DC) program is still non-convex, we employ the SCA method [18] that prescribes the iterative solution of non-convex problems by replacing the non-convex objective function and constraints with the suitable convex approximations. By denoting $P_A^k = \{P_A^k[n]\}_{n=1}^N$ and $P_U^k = \{P_U^k[n]\}_{n=1}^N$ as the set of transmit powers of Alice and the UAV at the $k$-th iteration and applying the first-order Taylor expansion [21], we can obtain the respective global upper bounds for the minus portion of the right-hand sides of (14) and (15) as

$$\log_2(1+a_3 P_A[n]+a_4[n] P_U[n]) \leq \log_2(1+a_3 P_A^k[n]+a_4[n] P_U^k[n])$$
$$+\frac{a_3(P_A[n]-P_A^k[n])}{1+a_3 P_A^k[n]+a_4[n] P_U^k[n]}+\frac{a_4[n](P_U[n]-P_U^k[n])}{1+a_3 P_A^k[n]+a_4[n] P_U^k[n]}. \tag{17}$$

Then, the problem (13) can be written as

$$\max_{P_{\mathrm{A}},P_{\mathrm{U}},S} \sum_{n=1}^{N} S[n], \tag{18}$$

$$\text{s.t.}\quad S[n] \leq \frac{B}{2}\{\log_2(1 + a_1 P_{\mathrm{A}}[n]) - a_{31}[n]P_{\mathrm{A}}[n] - a_{41}[n]P_{\mathrm{U}}[n] + a_6[n]\}\quad\text{for}\ \ n \in \mathcal{N}, \tag{19}$$

$$S[n] \leq \frac{B}{2}\{\log_2(1 + a_2 P_{\mathrm{A}}[n] + a_5[n]P_{\mathrm{U}}[n]) - a_{31}[n]P_{\mathrm{A}}[n] - a_{41}[n]P_{\mathrm{U}}[n] + a_6[n]\}$$
$$\text{for}\ \ n \in \mathcal{N}, \tag{20}$$

$$(4)\ \text{and}\ (11), \tag{21}$$

where $a_{31}[n] = a_3/(1 + a_3 P_{\mathrm{A}}^k[n] + a_4[n]P_{\mathrm{U}}^k[n])$, $a_{41}[n] = a_4[n]/(1 + a_3 P_{\mathrm{A}}^k[n] + a_4[n]P_{\mathrm{U}}^k[n])$, and $a_6[n] = \{(a_3 P_{\mathrm{A}}^k[n] + a_4[n]P_{\mathrm{U}}^k[n])/(1 + a_3 P_{\mathrm{A}}^k[n] + a_4[n]P_{\mathrm{U}}^k[n])\} - \log_2(1 + a_3 P_{\mathrm{A}}^k[n] + a_4[n]P_{\mathrm{U}}^k[n])$. The problem (18) is convex, and accordingly can be solved by the CVX solver [22]. Since the upper bounds in (17) suggests that any feasible solution $P_{\mathrm{A}}^k$ and $P_{\mathrm{U}}^k$ of (10) is also feasible for (18), the optimal value obtained by solving (18) serves as the lower bound for that of the problem (10).

### 3.2. Optimization of Power Splitting Ratio

The problem of optimizing the power splitting ratio $w$ with the fixed $P_{\mathrm{A}}$, $P_{\mathrm{U}}$, and $L_{\mathrm{U}}$ can be written by the definition of $R_{\mathrm{AB}}[n]$ in (6) as

$$\max_{w}\quad \sum_{n=1}^{N} R_{\mathrm{AB}}^w[n], \tag{22}$$

$$\text{s.t.}\quad (11), \tag{23}$$

where

$$R_{\mathrm{AB}}^w[n] = \frac{1}{2}\log_2\left\{1 + (g_{\mathrm{AB}}P_{\mathrm{A}}[n] + h_{\mathrm{UB}}[n]P_{\mathrm{U}}[n])\frac{w}{\sigma_{\mathrm{B}}^2}\right\}. \tag{24}$$

The problem (22) is convex, and accordingly can be solved by the CVX solver [22].

### 3.3. Optimization of UAV's Trajectory

In this subsection, given $P_{\mathrm{A}}$, $P_{\mathrm{U}}$, and $w$, we optimize $L_{\mathrm{U}}$ by introducing slack variables $\boldsymbol{m} = \{m[n]\}_{n=1}^{N}$, $\boldsymbol{q} = \{q[n]\}_{n=1}^{N}$, and $\boldsymbol{u} = \{u[n]\}_{n=1}^{N}$, with satisfying $m[n] \geq H^2 + \|L_{\mathrm{U}}[n] - L_{\mathrm{A}[n]}\|^2$, $q[n] \geq H^2 + \|L_{\mathrm{U}}[n] - L_{\mathrm{B}[n]}\|^2$, and $u[n] \leq H^2 + \|L_{\mathrm{U}}[n] - L_{\mathrm{E}[n]}\|^2$. Consequently, the problem (10) can be represented as

$$\max_{L_{\mathrm{U}},\boldsymbol{m},\boldsymbol{q},\boldsymbol{u}}\left[\frac{B}{2}\sum_{n=1}^{N}\left[\min\left\{\log_2\left(1 + \frac{b_1[n]}{m[n]}\right), \log_2\left(1 + \gamma_{\mathrm{AB}}[n] + \frac{b_2[n]}{q[n]}\right)\right\}\right.\right.$$

$$\left.\left. - \log_2\left(1 + \gamma_{\mathrm{AE}}[n] + \frac{b_3[n]}{u[n]}\right)\right]\right] \times \left(\frac{1}{\sum_{n=1}^{N} E_{\mathrm{P}}[n]}\right), \tag{25}$$

$$\text{s.t.}\quad m[n] \geq H^2 + \|L_{\mathrm{U}}[n] - L_{\mathrm{A}[n]}\|^2\quad\text{for}\ \ n \in \mathcal{N}, \tag{26}$$

$$q[n] \geq H^2 + \|L_{\mathrm{U}}[n] - L_{\mathrm{B}[n]}\|^2\quad\text{for}\ \ n \in \mathcal{N}, \tag{27}$$

$$u[n] \leq H^2 + \|L_{\mathrm{U}}[n] - L_{\mathrm{E}[n]}\|^2\quad\text{for}\ \ n \in \mathcal{N}, \tag{28}$$

$$(1)\ \text{and}\ (2), \tag{29}$$

where $b_1[n] = \rho_0 P_{\mathrm{A}}[n]/\sigma_{\mathrm{U}}^2$, $b_2[n] = w\rho_0 P_{\mathrm{U}}[n]/\sigma_{\mathrm{B}}^2$, and $b_3[n] = \rho_0 P_{\mathrm{U}}[n]/\sigma_{E}^2$.

To address the non-convexity of (25) due to the DC function depending on $\boldsymbol{L}_U$, we apply the SCA method [18] as in Section 3.1. By introducing slack variable $\boldsymbol{C} = \{C[n]\}_{n=1}^N$ for (25), we can rewrite the problem (25) as

$$\max_{\boldsymbol{L}_U,\boldsymbol{C},\boldsymbol{m},\boldsymbol{q},\boldsymbol{u}} \frac{\sum_{n=1}^N C[n]}{\sum_{n=1}^N E_P[n]} \tag{30}$$

$$\text{s.t.} \quad C[n] \leq \frac{1}{2}\left\{\log_2\left(1 + \frac{b_1[n]}{m[n]}\right) - \log_2\left(1 + \gamma_{AE}[n] + \frac{b_3[n]}{u[n]}\right)\right\} \quad \text{for } n \in \mathcal{N}, \tag{31}$$

$$C[n] \leq \frac{1}{2}\left\{\log_2\left(1 + \gamma_{AB}[n] + \frac{b_2[n]}{q[n]}\right) - \log_2\left(1 + \gamma_{AE}[n] + \frac{b_3[n]}{u[n]}\right)\right\} \quad \text{for } n \in \mathcal{N}, \tag{32}$$

$$(1), (2), (26), (27), \text{ and } (28). \tag{33}$$

By applying the first-order Taylor expansion [21], the first term in (31) with $\boldsymbol{m}^k = \{m^k[n]\}_{n=1}^N$ at the $k$-th iteration can be lower-bounded as

$$\log_2\left(1 + \frac{b_1[n]}{m[n]}\right) \geq \log_2\left(1 + \frac{b_1[n]}{m^k[n]}\right) - \frac{b_1[n](m[n] - m^k[n])}{\ln 2((m^k[n])^2 + b_1[n]m^k[n])}. \tag{34}$$

Similarly, the first log-term in constraint (32) with $\boldsymbol{q}^k = \{q^k[n]\}_{n=1}^N$ and the second log-terms in constraint (31) and (32) with $\boldsymbol{u}^k = \{u^k[n]\}_{n=1}^N$ in the $k$-th iteration can be lower-bounded as

$$\log_2\left(1 + \gamma_{AB}[n] + \frac{b_2[n]}{q[n]}\right) \geq \log_2\left(1 + \gamma_{AB}[n] + \frac{b_2[n]}{q^k[n]}\right)$$
$$- \frac{b_2[n](q[n] - q^k[n])}{\ln 2((1 + \gamma_{AB}[n])(q^k[n])^2 + b_2[n]q^k[n])} \tag{35}$$

and

$$\log_2\left(1 + \gamma_{AE}[n] + \frac{b_3[n]}{u[n]}\right) \geq \log_2\left(1 + \gamma_{AE}[n] + \frac{b_3[n]}{u^k[n]}\right)$$
$$- \frac{b_3[n](u[n] - u^k[n])}{\ln 2((1 + \gamma_{AE}[n])(u^k[n])^2 + b_3[n]u^k[n])}, \tag{36}$$

respectively. Next, we deal with the non-convexity of the helper UAV's flying energy consumption of $E_P[n]$ in (25). The terms for blade profile power and parasite power are convex, while the term for induced power is non-convex. To tackle with this issue, we also introduce the equivalent formulas of $E_P[n]$ with the slack variable $\boldsymbol{e} = \{e[n]\}_{n=1}^N$ as [10]:

$$\frac{1}{e^2[n]} \leq e^2[n] + \frac{v^2[n]}{v_0^2} \tag{37}$$

$$= e^2[n] + \frac{\|\boldsymbol{L}_U[n] - \boldsymbol{L}_U[n-1]\|^2}{v_0^2 t_s^2}, \tag{38}$$

with satisfying $e[n] \geq [(1 + (v[n])^4/(4v_0^4))^{\frac{1}{2}} - (v[n])^2/(2v_0^2)]^{\frac{1}{2}}$. By applying the first-order Taylor expansion [21], the right-hand side of (38) at any given points $\boldsymbol{e}^k = \{e^k[n]\}_{n=1}^N$ and $\boldsymbol{L}_U^k = \{\boldsymbol{L}_U^k[n]\}_{n=1}^N$ in the $l$-th iteration has the following lower bound:

$$e^2[n] + \frac{\|\boldsymbol{L}_U[n] - \boldsymbol{L}_U[n-1]\|^2}{v_0^2 t_s^2} \geq (e^k[n])^2 + 2e^k[n](e[n] - e^k[n]) - \frac{\|\boldsymbol{\psi}^k[n]\|^2}{v_0^2 t_s^2}$$
$$+ \frac{2}{v_0^2 t_s^2}(\boldsymbol{\psi}^k[n])^T(\boldsymbol{L}_U[n] - \boldsymbol{L}_U[n-1]) \triangleq F^k(e[n], \boldsymbol{L}[n], \boldsymbol{L}[n-1]), \tag{39}$$

where $\boldsymbol{\psi}^k[n] = \boldsymbol{L}_{\text{U}}^k[n] - \boldsymbol{L}_{\text{U}}^k[n-1]$. With (30)–(39), we can rewrite (25) as

$$\max_{\boldsymbol{L}_{\text{U}}, \boldsymbol{C}, \boldsymbol{m}, \boldsymbol{q}, \boldsymbol{u}, \boldsymbol{e}} \frac{\sum_{n=1}^{N} C[n]}{\sum_{n=1}^{N} \left[ P_0 \phi[n] + P_i e[n] + \frac{1}{2} d_0 \rho s A v^3[n] \right]} \tag{40}$$

$$\text{s.t. } C[n] \leq \frac{1}{2} \left\{ \log_2 \left( 1 + \frac{b_1[n]}{m^k[n]} \right) - \frac{b_1[n](m[n] - m^k[n])}{\ln 2 ((m^k[n])^2 + b_1[n] m^k[n])} \right.$$
$$\left. - \log_2 \left( 1 + \gamma_{\text{AE}}[n] + \frac{b_3[n]}{u^k[n]} \right) + \frac{b_3[n](u[n] - u^k[n])}{\ln 2 ((1 + \gamma_{\text{AE}}[n])(u^k[n])^2 + b_3[n] u^k[n])} \right\} \text{ for } n \in \mathcal{N}, \tag{41}$$

$$C[n] \leq \frac{1}{2} \left\{ \log_2 \left( 1 + \gamma_{\text{AB}}[n] + \frac{b_2[n]}{q^k[n]} \right) - \frac{b_2[n](q[n] - q^k[n])}{\ln 2 ((1 + \gamma_{\text{AB}}[n])(q^k[n])^2 + b_2[n] q^k[n])} \right.$$
$$\left. - \log_2 \left( 1 + \gamma_{\text{AE}}[n] + \frac{b_3[n]}{u^k[n]} \right) + \frac{b_3[n](u[n] - u^k[n])}{\ln 2 ((1 + \gamma_{\text{AE}}[n])(u^k[n])^2 + b_3[n] u^k[n])} \right\} \text{ for } n \in \mathcal{N}, \tag{42}$$

$$\frac{1}{e^2[n]} \leq F^k(e[n], \boldsymbol{L}[n], \boldsymbol{L}[n-1]) \text{ for } n \in \mathcal{N}, \tag{43}$$

$$e[n] \geq 0 \text{ for } n \in \mathcal{N}, \tag{44}$$

(1), (2), (26), (27) and (28). $\hspace{8cm}$ (45)

Since (40) is a fractional problem with the objective function consisting of a linear numerator and convex denominator along with the convex constraints (41)–(43), it can be optimally and efficiently solved by fractional programming techniques, such as the Dinkelbach method [19], which converts the original nonlinear fractional problem into a sequence of non-fractional problems to be solved by introducing an auxiliary variable until convergence.

### 3.4. Convergence and Complexity Analysis

The original problem (10) can be addressed by solving three subproblems alternatively, i.e., (18), (22), and (40). To sum up, we propose the efficient iterative alternating Algorithm 1 to obtain a high-quality local optimal solution of problem (10). In each iteration of the proposed Algorithm 1, three distinct subproblems are optimized as follows:

- One optimizes the transmit power of Alice $\boldsymbol{P}_{\text{A}}$ and transmit power of the UAV $\boldsymbol{P}_{\text{U}}$ with the fixed power splitting ratio $w$ and the UAV's trajectory $\boldsymbol{L}_{\text{U}}$;
- Another optimizes the power splitting ratio $w$ with fixed transmit power for Alice $\boldsymbol{P}_{\text{A}}$, transmit power for the UAV $\boldsymbol{P}_{\text{U}}$, and the UAV's trajectory $\boldsymbol{L}_{\text{U}}$;
- The other optimizes the UAV's trajectory $\boldsymbol{L}_{\text{U}}$ with fixed transmit power for Alice $\boldsymbol{P}_{\text{A}}$, transmit power for the UAV $\boldsymbol{P}_{\text{U}}$, and power splitting ratio $w$.

For formulating three subproblems (18), (22), and (40), the bounds in (17), (34)–(36), and (39) are adopted, by which we can obtain the optimal SEE value of the original problem (10). This approach based on those bounds ensures the convergence of the proposed Algorithm 1 as explained in [5]. Specifically, the proposed Algorithm 1 provides the local optimal solution of the problem (10) since the optimal solutions of three subproblems (18), (22), and (40) at each step are updated iteratively and alternatively to maximize the SEE with the inequality relationships of the bounds. The convergence of the proposed algorithm is verified numerically, as shown in Figure 2 as well. The proposed algorithm guarantees the convergence within about 15 iterations, where the larger mission time $T$ tends to achieve the smaller SEE due to the increase of flying energy consumption compared to the secrecy rate improvement.

For analyzing the computational complexity of the proposed Algorithm 1, we derive the computational complexity of the step to obtain the solution of the subproblems as mentioned above. For each iteration, the optimization (18) of the transmit powers can be solved

with the convex solver, whose computational complexity can be calculated as $\mathcal{O}(N^{3.5})$. In the case of optimizing Bob's power splitting ratio $w$ in (22), the computational complexity remains a constant, that is, $\mathcal{O}(1)$. When focusing on the optimization of the UAV trajectory as defined in (40), Algorithm 1 runs for the iterations of $Q_1 \times Q_2$, where the SCA algorithm's loop repeats $Q_1$ times, while the loop for Dinkelbach's algorithm repeats $Q_2$ times. Since there are the total of $6N$ variables in (40), the computational complexity for solving the subproblem (40) to attain the optimal UAV trajectory can be expressed as $\mathcal{O}(Q_1 Q_2 N^{3.5})$. Therefore, the total complexity of solving the original problem (10) is $\mathcal{O}(Q_1 Q_2 N^{3.5})$.

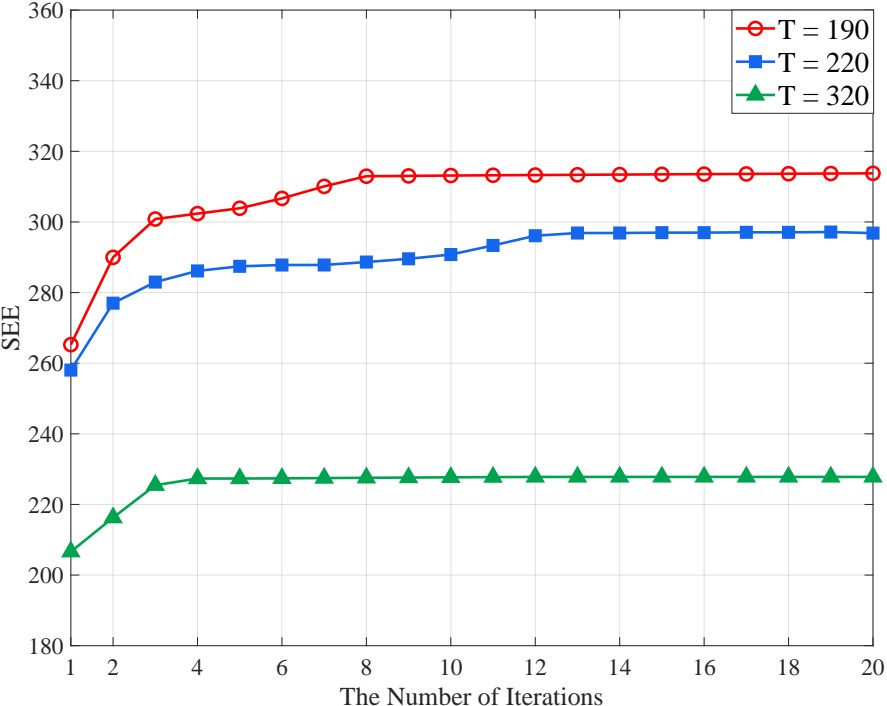

**Figure 2.** Convergence of the Algorithm 1.

## 4. Numerical Results

In this section, we evaluate the performance of the proposed Algorithm 1 via numerical experiments. For reference, we consider the following schemes.

1.  Optimal power and EH with linear UAV trajectory design (OptP&EH/LT) [23]: $\boldsymbol{P}_A$, $\boldsymbol{P}_U$, and $w$ are designed optimally by using the proposed Algorithm 1, while the linear UAV's trajectory is set by the following criterion:
    *   Insufficient mission time case ($T < (d_{\text{BU}_I} + \text{BU}_F)/v_{max}$): UAV flies from the initial spot to final spot with the constant velocity.
    *   Sufficient mission time case ($T \geq (d_{\text{BU}_I} + \text{BU}_F)/v_{max}$): UAV flies to Bob, and then turns to the final destination straightly with the constant velocity.
2.  Optimal UAV trajectory and EH with the equal power allocation (OptT&EH/EP) [11]: $\boldsymbol{L}_U$ and $w$ are optimized by using the proposed Algorithm 1 with fixed $P_A[n] = P_A^{\text{avg}}$ and $P_U[n] = P_U^{\text{avg}}$.
3.  Optimal EH with the equal power and the linear UAV trajectory (OptEH/EP&LT): $w$ is obtained by the proposed Algorithm 1 with $P_A[n] = P_A^{\text{avg}}$, $P_U[n] = P_U^{\text{avg}}$, and the linear UAV trajectory design.
4.  Optimal power and UAV trajectory without EH (OptP&T/NEH, Upperbound) [24]: $\boldsymbol{P}_A$, $\boldsymbol{P}_U$, and $\boldsymbol{L}_U$ are optimized by using the proposed Algorithm 1 without the EH operation at Bob.

The proposed algorithm, as referred to in Algorithm 1, consistently outperforms benchmark schemes in various scenarios. Remarkably, the performance of the proposed

algorithm approaches the upper bounds established by the OptP&T/NEH case, a testament to its effectiveness. Figures 3 and 4 illustrate the SEE as a function of mission duration $T$ and the optimal trajectory of the UAV determined by the proposed algorithm, respectively. These figures assume the scenario where Eve is positioned opposite Bob, resulting in less exposure. The SEE values achieved by all schemes display a concavity with respect to $T$, attributable to the power consumption's linear rate of increase surpassing the logarithmic rate of increase of the secrecy rate, as depicted in (9). Notably, the proposed algorithm performs better than all reference schemes except its upper bound of OptP&T/NEH case, with which it is nearly on par. Furthermore, in the case of the insufficient mission time, e.g., $T = 140$ to $180$ s, the UAV cannot afford to detour around Bob to maximize the SEE as demonstrated in the OptP&EH/LT and OptEH/EP&LT scenarios. Conversely, as the mission time $T$ extends, trajectory optimization (i.e., OptT&EH/EP) proves more effective than power optimization (i.e., OptP&EH/LT) in enhancing SEE for the scenario under consideration. Figure 4 shows the UAV hastening towards Bob and lingering within the allocated flight duration for optimal SEE before proceeding to the final destination. This behavior illustrates the relationship between mission time and SEE attainment: with limited time, the UAV's path is more direct, while additional time allows for more complex maneuvers to maximize SEE, highlighting the necessity of allowing adequate mission times for tasks demanding higher security levels in communication.

The coordinates of ground nodes are set as $L_A = (0, 0, 0)$, $L_{U_I} = (-100, 0, 20)$, $L_{U_F} = (300, 0, 20)$, and $L_B = (100, 200, 0)$. For Eve's location, we consider $L_E = (100, -200, 0)$ for the case when Eve is located at the opposite site of Bob, while $L_B = (100, 200, 0)$, $L_E = (200, 100, 0)$ for the case when Eve is located near Bob. We follow the UAV's system parameter setting of [10] and consider the remaining parameters as $H = 20$ m, $v_{max} = 3$ m/s, $\rho_0 = -60$ dB, $P_A^{avg} = 30$ dBm, $P_A^{peak} = 36$ dBm, $P_U^{avg} = 20$ dBm, $P_U^{peak} = 26$ dBm, $B = 1$ MHz, $\sigma_B^2 = \sigma_U^2 = \sigma_E^2 = -110$ dBm, $\eta = 0.7$, $\varphi = -90$ dBm, $\lambda = 10^{-4}$, and $\epsilon = 10^{-4}$. Figures 5 and 6 present the SEE as a function of mission time $T$ and the optimal UAV trajectory obtained by the proposed algorithm, respectively, when Eve is in close proximity to Bob, potentially compromising Bob's channel state information. As in Figure 3, the proposed algorithm is superior to the benchmark curves and comparable with the OptP&T/NEH case. Compared to Figure 3, due to the closer proximity of Eve, there is the possibility of the zero-valued SEE for the case with no power optimization, i.e., OptEH/EP&LT and OptT&EH/EP, when the given mission time is insufficient. The zero-valued SEE in scenarios without power optimization accentuates the gravity of power adjustments. While trajectory optimization is pivotal, without suitable power allocations, the UAV's communication can be rendered futile. With sufficient mission time, the UAV strategizes to maximize SEE by maneuvering away from Eve and towards Bob, as shown in Figure 6.

The juxtaposition between Figures 3 and 4 and Figure 5 and 6 elucidates the significance of Eve's location. When Eve is situated at the opposite side of Bob, the primary focus is on minimizing exposure. However, when Eve is in closer proximity to Bob, the strategy shifts towards a more aggressive approach, with the UAV flying closer to Bob while opposing Eve to ensure secure communication. This various nature underpins the necessity of the proposed optimal algorithms that can adjust not just based on the mission time but also based on potential threats.

We summarize the execution times in seconds (s) of benchmark schemes with the proposed algorithm as in Table 1. As in Table 1, although the proposed algorithm can achieve the best performance in terms of SEE, it is observed to take only 30~40% of the worst execution time of benchmark scheme, i.e, OptP&EH/LT under 140~180 s. For the sufficient mission time, since the UAV's trajectory optimization spends time to detour from the route to avoid the Eve, the proposed algorithm and benchmark scheme with the UAV's trajectory optimization take sufficient time, which is the trade-off to have the performance gain for SEE. The simulations were conducted on a machine equipped with an Intel Core i7-12700K CPU at 3.61 GHz, 32 GB of RAM, and a 1TB SSD. For the implementation of the proposed algorithm, we employed MATLAB R2022a.

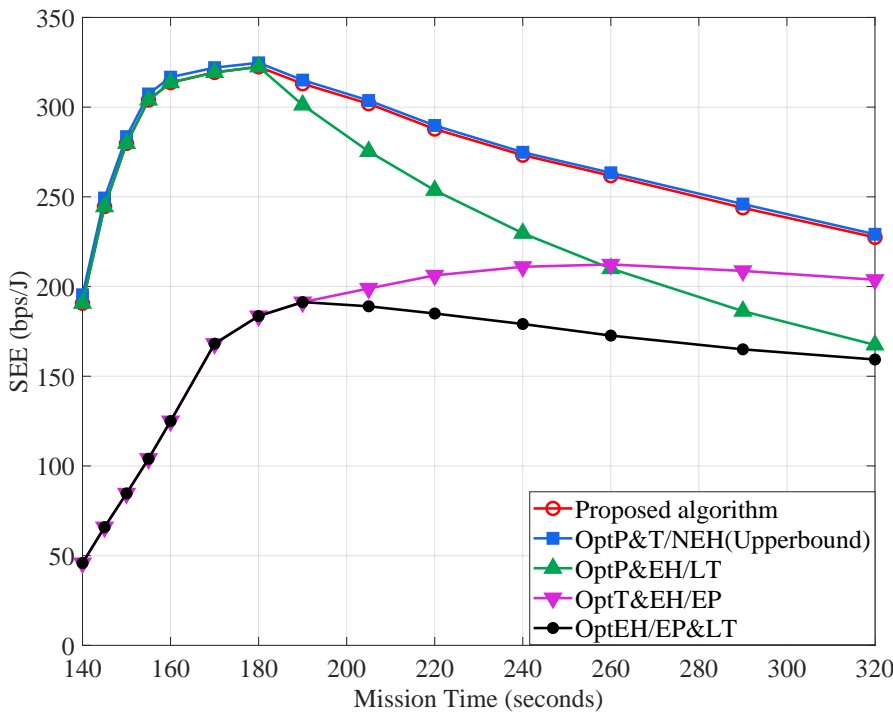

**Figure 3.** SEE of the UAV when Eve is located at the opposite site of Bob.

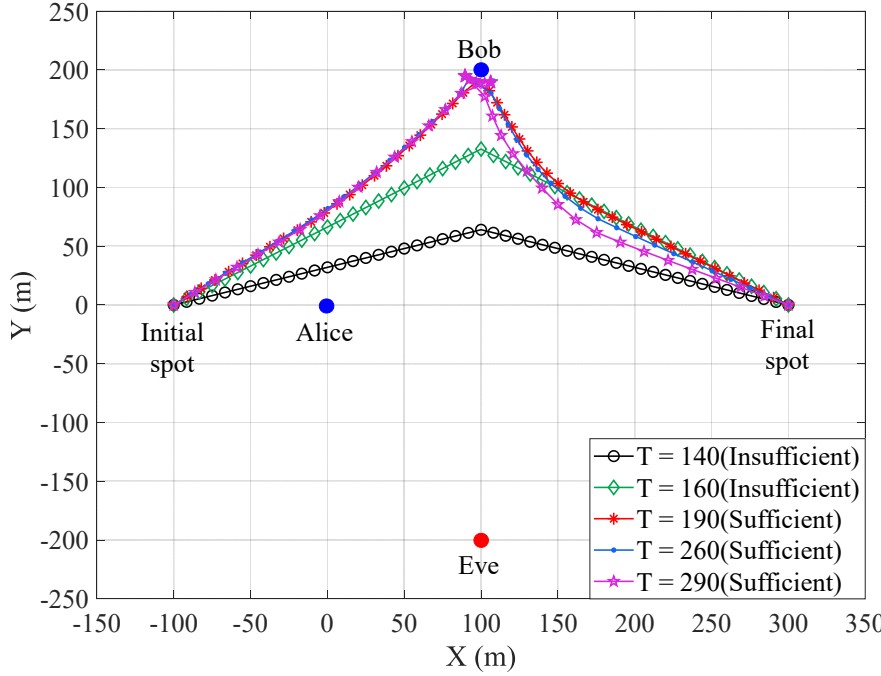

**Figure 4.** Optima l trajectory of the UAV when Eve is located at the opposite site of Bob.

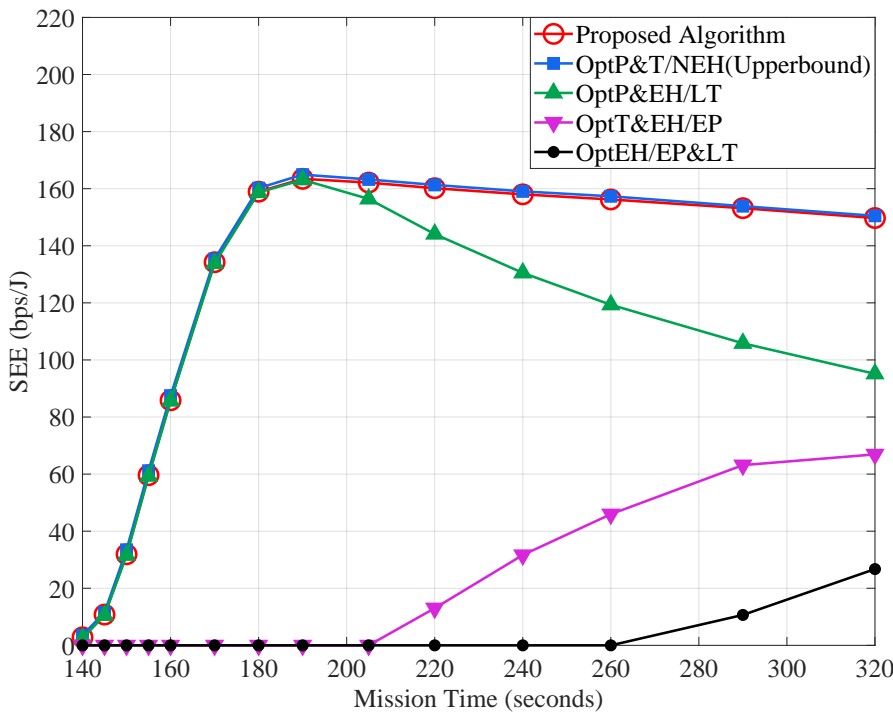

**Figure 5.** SEE of the UAV when Eve is located near Bob.

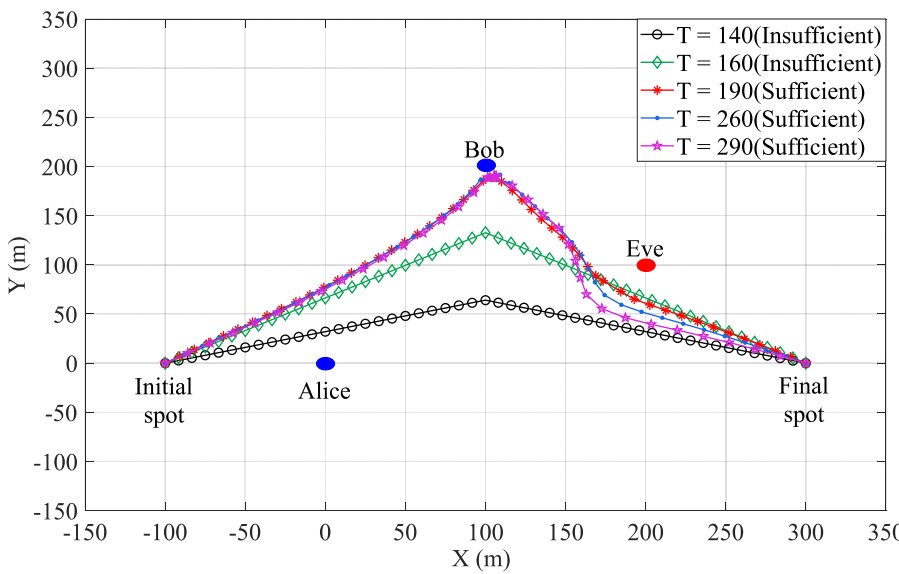

**Figure 6.** Optimal trajectory of the UAV when Eve is located near Bob.

**Table 1.** Execution times of benchmark schemes with the proposed algorithm.

| *T* (s) | 140 | 150 | 160 | 180 | 205 | 290 |
|---|---|---|---|---|---|---|
| Proposed Algorithm | 10.7 | 7.6 | 7.1 | 7.1 | 952.0 | 2721.0 |
| OptP&T/NEH [24] | 16.7 | 5.8 | 5.3 | 6.3 | 1047.8 | 2794.4 |
| OptP&EH/LT [23] | 23.5 | 20.3 | 19.1 | 24.5 | 13.8 | 12.6 |
| OptT&EH/EP [11] | 0.5 | 0.4 | 0.5 | 0.5 | 814.9 | 9906.4 |
| OptEH/EP&LT | 0.1 | 0.0 | 0.1 | 0.0 | 1.9 | 2.0 |

## 5. Concluding Remarks

In this work, we propose a secure UAV-assisted SWIPT communication system with the primary objective of maximizing the SEE. To achieve this, we address the joint SEE

maximization problem by simultaneously optimizing the transmit powers of Alice and the UAV, the power splitting ratio for EH operation at Bob, and the UAV's flying trajectory, all within the functional capability constraints of network devices. To tackle this optimization challenge, we have developed an efficient iterative alternating algorithm that combines SCA and Dinkelbach's method. The proposed algorithm, designed for UAV system environments, employs an offline-based optimization approach that ensures suitability and efficiency for mission planning and execution. This method significantly reduces computational complexities and enhances operational reliability, marking a significant advancement in the field of UAV communications. Performance validation through simulations confirms the superiority of our algorithm over conventional partial optimization schemes. Future work may consider scenarios involving multiple Bobs and UAVs supporting a variety of secure wireless communications. In conclusion, this research fills critical gaps in the existing literature on UAV-assisted systems. By introducing a novel approach, we believe our study establishes a new benchmark in the field. Our findings offer improved solutions and insights that can be crucial for future research, ensuring both energy efficiency and security in UAV-assisted communications.

**Author Contributions:** Conceptualization, D.H. and S.J.; methodology, D.H., S.J. and J.K.; project administration, J.K.; software, D.H. and J.K.; supervision, J.K.; validation, S.J., J.K. and J.K.; formal analysis, D.H., S.J. and J.K.; investigation, D.H.; resources, J.K.; data curation, D.H. and J.K.; writing—original draft preparation, D.H.; writing—review and editing, S.J., J.K. and J.K.; funding acquisition, J.K. All authors have read and agreed to the published version of the manuscript.

**Funding:** This research was supported by the MSIT (Ministry of Science and ICT), Korea, under the ITRC (Information Technology Research Center) support program (IITP-2020-0-01787) supervised by the IITP (Institute of Information & Communications Technology Planning & Evaluation). This work was supported by a National Research Foundation of Korea (NRF) grant funded by the Korean government (MSIT) (No. 2023R1A2C2005507) and an NRF grant funded by the MSIT (No. 2021R1F1A1050734).

**Data Availability Statement:** No new data were created or analyzed in this study. Data sharing is not applicable to this article.

**Conflicts of Interest:** The authors declare no conflict of interest.

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
