# Peer review of "Secrecy Energy Efficiency Maximization for Secure Unmanned-Aerial-Vehicle-Assisted Simultaneous Wireless Information and Power Transfer Systems"

_drones, doi:10.3390/drones7110672_

Round 1

Reviewer 1 Report

Comments and Suggestions for Authors

This paper proposes a joint SEE maximization technique for supporting secure UAV-assisted SWIPT communication systems. Through various experiments, it was confirmed that the proposed technique demonstrates superior performance compared to partial optimization methods. Despite being a straightforward approach, this paper is considered to yield quite meaningful research results when taking into account the complexity of the problem this study aims to address. To enhance the completeness of the paper, it would be beneficial to address the following points:

1. What are the advantages of the proposed technique compared to the benchmark method OptP&T/NEH?

2. The increase in computational complexity due to joint optimization and the resulting processing time delays can be critical issues in the UAV communication environment. It would be beneficial to include discussions on this matter in the paper.

Reviewer 2 Report

Comments and Suggestions for Authors

The paper introduces a secrecy energy efficiency maximization problem formulate model and solution algorithm for the confidential message to an energy-constrained device in the presence of a passive eavesdropper. The simulation verified the proposed algorithm when compared to other optimization schemes. However, there are some specific issues that should be addressed:

1. To provide a comprehensive evaluation, it would be beneficial to include a separate table comparing the execution times of baseline schemes with the proposed algorithm.

2. What is the SEE unit in y-axis and the unit of T in Figure 2, what is the performance different between the different T?

3. It is advisable to expand the discussion of Figure 3,4,5,6 section, providing a more comprehensive comparison and effectiveness of propose model and algorithm.

4. What is the result if the Eve is between the Alice and Bob in optimal trajectory discussion?

Comments on the Quality of English Language

Minor editing of English language required

Reviewer 3 Report

Comments and Suggestions for Authors

The main contribution of this paper is to propose a secure UAV-assisted SWIPT system that uses an UAV acting as a helper node to improve the system's SEE as well as to aid the energy harvesting(EH) of battery-limited ground device following the SWIPT protocol. Via simulations, the feasibility and the effectiveness of the proposed algorithm are corroborated compared to the conventional partial optimizations.

However, there are still some problems with the article as follows:

1. There are few references to the latest literature, and the references in this paper are basically from before 2020.

2. It is suggested that the comparison with other methods should be increased.

Based on the above issues: the authors are advised to revise and re-review.

Comments on the Quality of English Language

The main contribution of this paper is to propose a secure UAV-assisted SWIPT system that uses an UAV acting as a helper node to improve the system's SEE as well as to aid the energy harvesting(EH) of battery-limited ground device following the SWIPT protocol. Via simulations, the feasibility and the effectiveness of the proposed algorithm are corroborated compared to the conventional partial optimizations.

However, there are still some problems with the article as follows:

1. There are few references to the latest literature, and the references in this paper are basically from before 2020.

2. It is suggested that the comparison with other methods should be increased.

Based on the above issues: the authors are advised to revise and re-review.

Reviewer 4 Report

Comments and Suggestions for Authors

As a reviewer, I have some concerns about the paper "Secrecy Energy Efficiency Maximization for Secure UAV-Assisted SWIPT Systems." The following are some concerns:

In terms of the recommended strategy, the study does not present enough evidence of uniqueness. The authors mention a "secure UAV-assisted SWIPT system," but the approach's originality is not explicitly articulated or demonstrated. The paper should include a more in-depth assessment of comparable work as well as an explanation of how their work advances the state-of-the-art.

The problem statement is unclear and poorly defined. The paper presents the notion of secrecy energy efficiency (SEE), but does not define it properly or provide a concrete issue formulation. The absence of a well-defined problem makes it difficult for readers to understand the paper's objectives and contributions.

These authors provided many references to support a small argument, such as [5]-[15], [5]-[10], and so on. This is simply throwing the bundle of articles into the air and then challenging the reader to discover something of relevance among the resultant jumble. It does not assist the reader in locating additional information or evidence. Rather, demonstrate your understanding of the literature by directing the reader to one or two publications that clearly clarify your ideas. It should be avoided.

The study lacks essential technical details that readers would need to understand and replicate the proposed method. The algorithm used to solve the optimization problem is briefly stated, however the publication does not go into detail about its implementation, assumptions, or mathematical foundations.

The paper does not include enough references to back up its assertions and lay a solid foundation for the research. When addressing conventional methods and optimizations, it is critical to incorporate references to past work in related domains. The report also has weak wording and a lack of clarity.

Several grammatical and structural difficulties must be addressed in order to improve the paper's readability and comprehension.

In short, according to the journal scope, this article work is very limited.

Comments on the Quality of English Language

Extensive editing of English language required

Round 2

Reviewer 3 Report

Comments and Suggestions for Authors

The revised version addressed my questions.